# Bioassay-Guided Fractionation with Antimalarial and Antimicrobial Activities of *Paeonia officinalis*

**DOI:** 10.3390/molecules27238382

**Published:** 2022-12-01

**Authors:** Mamdouh Nabil Samy, Basma Khalaf Mahmoud, Nourhan Hisham Shady, Usama Ramadan Abdelmohsen, Samir Anis Ross

**Affiliations:** 1Department of Pharmacognosy, Faculty of Pharmacy, Minia University, Minia 61519, Egypt; 2National Center for Natural Products Research, School of Pharmacy, University of Mississippi, Oxford, MS 38677, USA; 3Department of Pharmacognosy, Faculty of Pharmacy, Deraya University, New Minia 61111, Egypt; 4Department of BioMolecular Science, Division of Phamacognosy, School of Pharmacy, University of Mississippi, Oxford, MS 38677, USA

**Keywords:** *Paeonia officinalis*, phytoconstituents, antimalarial, antimicrobial

## Abstract

Bioassay-guided fractionation technique of roots of *Paeonia officinalis* led to isolation and structure elucidation of seven known compounds, including four monoterpene glycosides: lactiflorin (**1**), paeoniflorin (**4**), galloyl paeoniflorin (**5**), and (*Z*)-(1*S*,5*R*)-*β*-pinen-10-yl *β*-vicianoside (**7**); two phenolics: benzoic acid (**2**) and methyl gallate (**3**); and one sterol glycoside: *β*-sitosterol 3-*O*-*β*-D-glucopyranoside (**6**). The different fractions and the isolated compounds were evaluated for their antimicrobial and antimalarial activities. Fraction II and III showed antifungal activity against *Candida neoformans* with IC_50_ values of 28.11 and 74.37 µg/mL, respectively, compared with the standard fluconazole (IC_50_ = 4.68 µg/mL), and antibacterial potential against *Pseudomonas aeruginosa* (IC_50_ = 20.27 and 24.82 µg/mL, respectively) and *Klebsiella pneumoniae* (IC_50_ = 43.21 and 94.4 µg/mL, respectively), compared with the standard meropenem (IC_50_ = 28.67 and 43.94 µg/mL, respectively). Compounds **3** and **5** showed antimalarial activity against *Plasmodium falciparum* D6 with IC_50_ values of 1.57 and 4.72 µg/mL and *P. falciparum* W2 with IC_50_ values of 0.61 and 2.91 µg/mL, respectively, compared with the standard chloroquine (IC_50_ = 0.026 and 0.14 µg/mL, respectively).

## 1. Introduction

Malaria is one of the major global life-threatening diseases, especially among children and pregnant women [1]. Malaria is caused by five protozoan species of genus *Plasmodium*, called *Plasmodium falciparum, P. vivax, P. malariae, P. ovale, and P. knowlesi* [2]. The diversity and ability of *P. falciparum* to evade the host immune responses make it the most deadly form of malaria [2,3], and, therefore, the WHO recognizes the importance of malaria as a major cause of morbidity and mortality, particularly in sub-Saharan Africa [4]. *P. falciparum* spread is caused by *Anopheles gambiae* and *An. Funestus*, which are the highly prevalent mosquitoes [1]. The natural products application in therapeutic management against microorganism-caused diseases presents advantages that place them above drugs derived from synthetic sources. Owing to the comparably minimal side effects of these drugs, their toxicological and pharmacological activity, and likewise gastroenterology and bacteriology, revealed remarkable interest in the natural products’ pharmacological activities against different infectious agents. Therefore, the natural products are capable of aiding the world health system in the discovery of new effective drugs to overcome the emergency of microbial resistance to available drugs [5,6].

*Paeonia* genus (family Paeoniaceae) consists of approximately of 33 species, which are mainly distributed in warm temperate regions of Europe and Asia, among them a total of 11 species are found in southwest and northwest China. In traditional Chinese medicine, *Paeonia* roots are considered members of the most important crude drugs, whereas they are applied as analgesic, sedative, and anti-inflammatory agents and as remedies for cardiovascular, stagnated blood, and female genital diseases. Monoterpene glycosides are the predominant class of phytoconstituents in *Paeonia* genus, together with triterpenoids, flavonoids, phenols, and tannins [7,8,9].

*Paeonia officinalis* (European peony, peony) is a perennial herb native to south-eastern Europe and has been widely introduced as a garden plant [10]. *P. officinalis* has tuberous fleshy roots and a stout, erect, branched, glabrous stem. The leaves are ternate or biternate and have ovate lanceolate segments, with a dark green color above and lighter below. *P. officinalis* roots has been included for years in the systems of medicine in Unani, Ayurvedic, and Homeopathic, together with Indian and Chinese systems of medicines [10,11]. *P. officinalis* roots are included as a component of several antioxidant preparations in Unani medicine, while in the Ayurvedic medicine, it represents a main part of many medicinal formulations to treat several diseases, such as jaundice, hepatitis, hepatomegaly, liver dysfunction, and cirrhosis [12]. The literature survey revealed the wide range of safety of the aqueous extracts of *P. officinalis*, in which it did not cause any mortality up to 2000 mg/kg, it also has a high protection level against CCl_4_ toxicity [10], along with its strong antioxidant potential through several in vitro assays by using Folin–Ciocalteu, 2,2-diphenyl-1-picrylhydrazyl radical (DPPH), 2,2′-azino-*bis*-3-ethylbenzothiazoline-6-sulfonic acid (ABTS), oxygen radical absorbance capacity (ORAC), hydroxyl radical antioxidant capacity (HORAC), hydroxyl radical scavenging capacity (HOSC), and cellular antioxidant activity (CAA) assays [8]. The roots contain asparagin, benzoic acid, flavonoids, paeoniflorin, paeonin, paeonol, protoanemonin, tannic acid, triterpenoids, and volatile oil [13,14,15]. The present study aims to discuss the isolation and identification of different phytoconstituents, in addition to their evaluation as antimalarial and antimicrobial agents.

## 2. Results and Discussion

The use of VLC technique fractionation of the EtOAc fraction of ethanolic extract of *P. officinalis* by using the gradient elution of DCM-MeOH resulted in four subfractions Fraction I–IV. The different subfractions were evaluated as antimalarial and antimicrobial agents (Table 1 and Table 2), where fractions II and III showed antifungal activity against *Candida neoformans* with IC_50_ values of 28.11 and 74.37 µg/mL, respectively, compared with the standard fluconazole (IC_50_ = 4.68 µg/mL), and antibacterial potential against *Pseudomonas aeruginosa* (IC_50_ = 20.27 and 24.82 µg/mL, respectively) and *Klebsiella pneumoniae* (IC_50_ = 43.21 and 94.4 µg/mL, respectively), compared with the standard meropenem (IC_50_ = 28.67 and 43.94 µg/mL, respectively), revealing the potency of the two subfractions as antimalarial and antimicrobial drugs.

According to the obtained biological guided assay, the two subfractions II and III were subjected to different chromatographic techniques as silica gel and Sephadex LH-20 column chromatography in order to purify the secondary metabolites, which are responsible for the subfractions bioactivity, leading to the isolation and structural elucidation of seven compounds (Figure 1), which were identified as lactiflorin (**1**) [16], benzoic acid (**2**) [17], methyl gallate (**3**) [18], paeoniflorin (**4**) [19], galloyl paeoniflorin (**5**) [20], *β*-sitosterol 3-*O*-*β*-D-glucopyranoside (**6**) [21], and (*Z*)-(1*S*,5*R*)-*β*-pinen-10-yl *β*-vicianoside (**7**) [22]. The structures of the isolated compounds were elucidated using different spectroscopic analyses, such as 1D and 2D NMR experiments (^1^H, ^13^C, DEPT, COSY, HMQC, and HMBC), as well as HR-ESI-MS analysis (Appendix A). Compounds **1, 3,** and **5–7** were reported for the first time in this plant.

The antimalarial and antimicrobial evaluation of the identified compounds revealed the potency of compounds **3** and **5** against *Plasmodium falciparum* D6 with IC_50_ values of 1.57 and 4.72 µg/mL and *P. falciparum* W2 with IC_50_ values of 0.61 and 2.91 µg/mL, respectively, comparing with the standard chloroquine (IC_50_ = 0.026 and 0.14 µg/mL, respectively), with no exhibition of antimicrobial activity.

The root extracts of *P. officinalis* have been utilized in Indian and Chinese medicine for a long time due to their pharmacological effects, which include neuroprotection, antihypertensive, and anti-ulcer [11]. Ethanol extracts of *P. officinalis* petals demonstrated antioxidant and antibacterial activities toward *Staphylococcus aureus* and *Escherichia coli* [23]. In the present study, fractions II and III showed antifungal activity against *C. neoformans* and antibacterial potential against *P. aeruginosa* and *K. pneumoniae.* The isolated compounds were inactive against the assayed microorganisms at the tested concentration.

In the antimalarial assays, fractions II and III demonstrated low activity against *P. falciparum* D6 and W2. Methyl gallate (**3**) and galloyl paeoniflorin (**5**) showed remarkable antiplasmodial activity against the *P. falciparum* D6 and W2 (Table 2). The antimalarial potency of the isolated compounds, was attributable to the presence of the number of phenolic hydroxyl groups in the galloyl moiety, indicated its role in the inhibition of *Plasmodium*. Fractions II and III and the isolated compounds did not exhibit cytotoxicity toward mammalian kidney fibroblasts (Vero).

## 3. Materials and Methods

### 3.1. General Experimental Procedures

Proton (^1^H) and ^13^C NMR spectra were recorded on Bruker Avance 400 MHz instrument. High resolution-electrospray ionization-mass spectrum (HR-ESI-MS) was taken on a LTQ Orbitrap XL mass spectrometer. Solvents used in this work, e.g., petroleum ether (pet. ether; B.p. 60–80 °C), dichloromethane (DCM), ethyl acetate (EtOAc), methanol (MeOH), and ethanol (EtOH), were purchased from Fisher Scientific, USA. Deuterated solvents (Sigma-Aldrich, Darmstadt, Germany), including methanol (CD_3_OD) and pyridine (C_5_D_5_N), were used for nuclear magnetic resonance (NMR) spectroscopic analyses. Column chromatography (CC) was performed using silica gel 60 (El-Nasr Company for Pharmaceuticals and Chemicals, Egypt; 60–120 mesh) or Sephadex LH-20 (0.25–0.1 mm, GE Healthcare, Sweden), while silica gel GF_254_ for thin layer chromatography (TLC) (0.25–0.1 µm, El-Nasr Company for Pharmaceuticals and Chemicals, Egypt) was employed for vacuum liquid chromatography (VLC) (6 × 30 cm, 90 g) at room temperature, and then the sample was loaded as solute and the elution was produced by the aid of water vacuum pump.

Thin layer chromatography (TLC) analyses were carried out using pre-coated silica G plates w/UV254 (Sorbent Technologies, Norcross, GA, USA; 20 × 20 cm, 200 µm in thickness). Ultraviolet lamp (UVP, LLC, Spectroline, Westbury, NY, USA) was used for visualization of spots on thin layer chromatograms at 254 and/or 365 nm. Spots were visualized by spraying with 2% vanillin–sulfuric acid in ethanol followed by heating at 110 °C on a hot plate.

### 3.2. Plant Material

The roots of *P. officinalis* were collected in January 2021 from public nurseries, Minia governorate, Egypt. Authentication of the plant was identified by Prof. Dr. Nasser Barakat, Department of Botany, Faculty of Science, Minia University, Minia, Egypt. A voucher specimen (Mn-ph-Cog-062) has been deposited in the Herbarium of Pharmacognosy Department, Faculty of Pharmacy, Minia University, Minia, Egypt.

### 3.3. Extraction and Isolation

The air-dried roots of *P. officinalis* (650 g) were powdered and extracted through maceration with 95% ethanol at room temperature and concentrated under reduced pressure, affording 60 g of solvent-free residue. The residue was suspended in 50 mL of distilled water to perform liquid–liquid fractionation, and then defatted with pet. ether, followed by partitioning with EtOAc (100 mL each × 6). The solvents were separately evaporated under vacuum, affording 4.0 g of pet. ether fraction and 25 g of EtOAc fraction. Finally, the remaining mother liquor was concentrated under reduced pressure to afford aqueous fraction.

The EtOAc fraction of *P. Officinalis* roots was fractionated by using VLC technique, in which it is eluted initially with DCM and then the polarity was increased gradually in 10% by MeOH until DCM-MeOH was 70:30. Each polarity was collected and concentrated under reduced pressure, affording four subfractions (E-1:E-4).

The subfraction II (7.0 g), which was eluted by DCM-MeOH (90:10), was further subjected to fractionation using silica gel CC (76 × 3 cm, 210 g), using DCM-MeOH gradient mixtures in order to increase the polarity gradually to 2% by MeOH until DCM-MeOH was 80:20, then the column washed by MeOH. The effluents were collected in fractions and concentrated under reduced pressure to give 31 subtractions. Subfraction II-12 (106.8 mg) was purified through Sephadex LH-20 CC using MeOH, yielding compounds **1** (23.9 mg), **2** (16.8 mg), and **3** (92.8 mg). Subfraction II -20 (1.60 g) was purified through Sephadex LH-20 CC using MeOH, yielding compound **4** (25.8 mg). Subfraction II-29 (414.0 mg) was purified through Sephadex LH-20 CC using MeOH, yielding compound **5** (25.8 mg).

The subfraction III (13.0 g), which was eluted by DCM-MeOH (80:20), was further fractionated on silica gel CC (77 × 4 cm, 390 g), using DCM-MeOH gradient mixtures in order to increase the polarity gradually to 2% by MeOH until DCM-MeOH was 80:20, then the column washed by MeOH. The effluents were collected in fractions and concentrated under reduced pressure to give 34 subtractions. Subfraction III-8 produced compound **6** (38.8 mg). Subfraction III-20 (370.3 mg) was purified through Sephadex LH-20 CC using MeOH, yielding compound **7** (45.1 mg).

### 3.4. Evaluation of Antimicrobial Activity

The microorganism strains that were employed to test the antimicrobial activity were *Candida albicans* Pinh, *C. neoformans* Pinh, *Aspergillus fumigatus* Pinh, methicillin-resistant *Staphylococcus aureus* Pinh (MRS), *E. coli* Pinh, *Pseudomonas aeruginosa* Pinh, *Klebsiella pneumoniae* Pinh, and *Vancomycin-resistant Enterococci* (VRE) Pinh. The strains were bought from the American Type Culture Collection. A modified version of the CLSI methodology was used for the susceptibility testing. A final DMSO concentration of 1% was maintained in the assay by serially diluting all samples in 20% DMSO/saline and transferring them in duplicate to 384 well flat-bottom microplates. Following the McFarland standard, inocula were created by adjusting the OD_630_ of microbe suspensions in incubation broth. *C. albicans* was cultured in RPMI 1640 broth (2% dextrose, 0.03% glutamine, and MOPS at pH 6); *C. neoformans* in Sabouraud Dextrose; MRS, VRE, *E. coli*, *K. pneumoniae*, and *P. aeruginosa* in cation-adjusted Mueller–Hinton at pH 7.3, and A. fumigatus in RPMI 1640 broth (2% dextrose, 0.03% glutamine, buffered with 0.165 M MOPS at pH 7). Five percent Alamar Blue was added in *A. fumigatus*, VRE, and MRS. In every assay, there were drug controls for bacteria and fungi. *A. fumigatus* and *C. albicans* were incubated at 35 °C for 48 h, while *C. neoformans* was cultured at 35 °C for 68 to 72 h. MRS, VRE, *E. coli*, *K. pneumoniae*, and *P. aeruginosa* were also incubated at this temperature for 18 to 24 h at 35 °C. A Bio-Tek plate reader was used to measure the optical density (530 nm) or fluorescence (544ex/590em) before and after incubation [24,25].

### 3.5. Evaluation of Antimalarial Activity

The assay is based on the measurement of plasmodial LDH activity. The test was conducted using a suspension of red blood cells infected with D6 or W2 strains of *P. falciparum* (200 µL, with 2% parasitemia, and 2% hematocrit in RPMI 1640 medium supplemented with 10% human serum and 60 µg/mL Amikacin) into the wells of a 96-well plate containing 10 µL of test samples diluted in medium at various concentrations. The plate was put in a modular incubation chamber (Billups-Rothenberg, San Diego, CA, USA); flushed with a gas mixture consisting of 90% N_2_, 5% O_2_, and 5% CO_2_; and incubated for 72 h at 37 °C. Parasitic LDH activity was measured by utilizing Malstat reagent (Flow Inc., Portland, OR, USA). Briefly, 20 µL of the incubation mixture was combined with 100 µL of the Malstat reagent and incubated at room temperature for 30 min. Then, twenty microliters of a 1:1 mixture of NBT/PES (Sigma, St. Louis, MO, USA) was added and the plate was further incubated in the dark for 1 h. The reaction was then stopped by the addition of 100 µL of a 5% acetic acid solution. The plate was read at 650 nm using the EL-340 Biokinetics Reader (Bio-Tek Instruments, Winooski, VT, USA). The dose–response curves were used to calculate IC_50_ values. Chloroquine was used in each experiment as the reference drug. Dimethyl sulfoxide (0.25%) was used as vehicle control [26].

## 4. Conclusions

The biological-based fractionation of *P. officinalis* roots resulted in the isolation and structure elucidation of seven compounds, including four monoterpene glycosides, two phenolics, and one sterol glycoside, in which compounds **3** and **5** showed antimalarial activity against *Plasmodium falciparum* D6 and *P. falciparum* W2. On the other hand, no other compound showed either antifungal or antibacterial activity. Accordingly, *P. officinalis* is considered to be a dietary supplement that exhibited antimicrobial and antimalarial activities with no apparent cytotoxicity against mammalian cells.

## Figures and Tables

**Figure 1 molecules-27-08382-f001:**
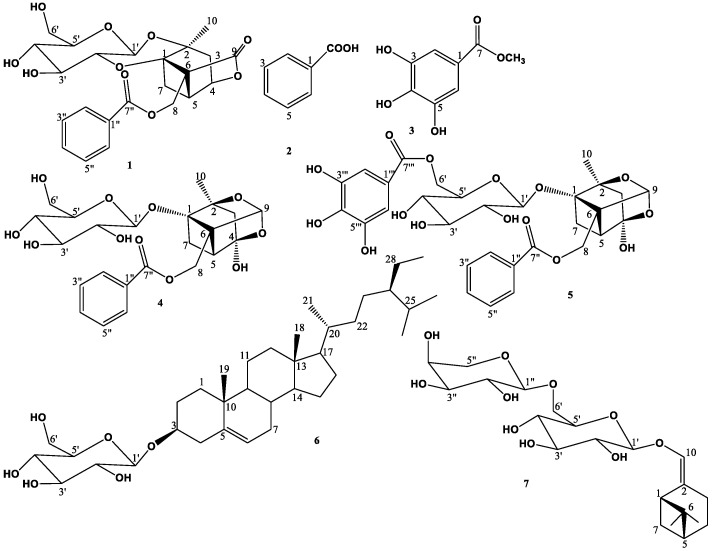
Structures of the isolated compounds of *P. officinalis* roots.

**Table 1 molecules-27-08382-t001:** IC_50_ (µg/mL) of the antimicrobial activity of fractions of *P. officinalis*.

	*C. neoformans*	MRS	*P. aeruginosa*	*K. pneumoniae*
Fraction II	28.106	194.869	20.216	43.214
Fraction III	74.372	>200	24.716	94.405
Fluconazole	4.684	-	-	-
Meropenem	-	46.921	28.542	9.143

**Table 2 molecules-27-08382-t002:** IC_50_ (µg/mL) of the antiplasmodial activity of fractions and isolated compounds from *P. officinalis*.

	*P. falciparum* D6	SI	*P. falciparum* W2	SI	VERO
Methyl gallate (**3**)	1.57	>3	0.61	>7.8	>4.76
Galloyl paeoniflorin (**5**)	4.72	>1	2.91	>1.6	>4.76
Fraction II	19.48	>2.4	8.06	>5.9	>47.60
Fraction III	24.57	>1.9	15.51	>3.1	>47.60
Chloroquine	0.026	>9	0.14	>1.8	>0.24

SI: Selectivity index.

## Data Availability

Not applicable.

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
