# Peer review of "Bioassay-Guided Fractionation with Antimalarial and Antimicrobial Activities of Paeonia officinalis"

_molecules, 2022, doi:10.3390/molecules27238382_

Round 1

Reviewer 1 Report

Dear Authors

Regarding to the  Manuscript ID: molecules-2055515

Title: Bioassay-guided fractionation and antimalarial and antimicrobial activities of Paeonia officinalis

Kindly have a look to all the comments provided in the attached manuscript

Also it is better to add  Figures;  S4, S7, S11, S15, S16 ,S19 and  S26 from the supplementary file to the results and discussions

Best regards

Author Response

Kindly have a look to all the comments provided in the attached manuscript.

  • All the comments provided in the attached manuscript were corrected

Also it is better to add  Figures;  S4, S7, S11, S15, S16 ,S19 and  S26 from the supplementary file to the results and discussions

  •  we added citation of supplementary figures to results and discussions

Reviewer 2 Report

The emphasis of this manuscript is on the antimalarial activity of fractions and individual compounds from Paeonia officinalis root extract. The literature survey showed no results for such studies regarding this plant species. Phytochemical study is correctly described. After reviewing, my comments concerning the manuscript are general positive, but needs some minor revision prior to acceptance.

# Table 1 presents the results for antimicrobial activity of fractions but not of individual compounds. Please correct the table title and include the designation and measure units of IC50.

# Table 2 presents the results of the antiplasmodial activity of fractions but not of the extract. Please correct the table title and add the designation and measure units of IC50.

# I would recommend adding in Section Introduction the name of the plant family to which the genus Paeonia belongs.

Author Response

Table 1 presents the results for antimicrobial activity of fractions but not of individual compounds. Please correct the table title and include the designation and measure units of IC50.

  • Title of table 1 was corrected 

Table 2 presents the results of the antiplasmodial activity of fractions but not of the extract. Please correct the table title and add the designation and measure units of IC50.

  • Title of table 2 was corrected 

I would recommend adding in Section Introduction the name of the plant family to which the genus Paeonia belongs.

  • The name of the family was in the introdcution 

Reviewer 3 Report

The manuscript entitled “Bioassay-guided fractionation with antimalarial and antimicrobi-2 al activities of Paeonia officinalis”, submitted to the journal presents a laboratory study aiming isolation and identification of different phytoconstituents of Paeonia officinalis roots and evaluation their antimalarial antimicrobial activity potential. The topic is current in connection with the growing importance of the search for new natural medicinal substances.

I have the following comments on the article:

-          The results of the antimicrobial and antimalarial activity tests are mentioned only in the text. They must be presented in tabular or other form in the article or supplement.

-          The discussion of the results is weak, they should be more compared with the results described in the literature.

-          The spectra of substances identified in the extracts (rows 176 - 230) should be given in the supplement.

-          The conclusion should answer the aim of this study. To conclude what new findings were found and how the findings advance our knowledge in the field. The antimicrobial potential of fractions II and III should be assessed.

 On my opinion, manuscript could be accepted for the publication after major revision.

Author Response

The results of the antimicrobial and antimalarial activity tests are mentioned only in the text. They must be presented in tabular or other form in the article or supplement.

  • the results were presented in tables 1 and 2

The discussion of the results is weak, they should be more compared with the results described in the literature.

  • The discussion was improved

         The spectra of substances identified in the extracts (rows 176 - 230) should be given in the supplement.

  •  The spectra of substances identified were moved to supplementary materials 

         The conclusion should answer the aim of this study. To conclude what new findings were found and how the findings advance our knowledge in the field. The antimicrobial potential of fractions II and III should be assessed.

  • The conclusion was improved

Round 2

Reviewer 3 Report

The authors have satisfactorily implemented my comments, I wish them much success in their future work.